# Pedunculopontine Gamma Band Activity and Development

**Edgar Garcia-Rill [1],\*, Brennon Luster [1], Susan Mahaffey [1], Melanie MacNicol [1], James R. Hyde [1], Stasia M. D'Onofrio [1] and Cristy Phillips [2]**

[1] Center for Translational Neuroscience, Department of Neurobiology and Developmental Sciences, University of Arkansas for Medical Sciences, Little Rock, AR 72205, USA;
E-Mails: BRLuster@uams.edu (B.L.); SCMahaffey@uams.edu (S.M.); macnicolmelanie@uams.edu (M.M.); JRHyde@me.com (J.R.H.); sdonofrio@uams.edu (S.M.D.O.)
[2] Department of Physical Therapy, Arkansas State University, Jonesboro, AR 72476, USA;
E-Mail: cphillips@astate.edu

\* Author to whom correspondence should be addressed; E-Mail: GarciaRillEdgar@uams.edu;
Tel.: +1-501-686-5167; Fax: +1-501-526-7928.

**Abstract:** This review highlights the most important discovery in the reticular activating system in the last 10 years, the manifestation of gamma band activity in cells of the reticular activating system (RAS), especially in the pedunculopontine nucleus, which is in charge of waking and rapid eye movement (REM) sleep. The identification of different cell groups manifesting P/Q-type $Ca^{2+}$ channels that control waking *vs.* those that manifest N-type channels that control REM sleep provides novel avenues for the differential control of waking *vs.* REM sleep. Recent discoveries on the development of this system can help explain the developmental decrease in REM sleep and the basic rest-activity cycle.

**Keywords:** basic rest-activity cycle; $Ca^{2+}$ channels; connexin 36; electrical coupling; parafascicular; preconscious awareness; REM sleep; reticular activating system; subcoeruleus; waking

## 1. Introduction

In this review, we will focus on implications of the discovery that every recorded cell in critical reticular activating system (RAS) centers, especially the pedunculopontine nucleus (PPN), manifests

beta/gamma band oscillations mediated by intrinsic membrane properties [1,2]. In fact, this property is endemic to PPN neurons irrespective of transmitter type, firing pattern, or electrophysiological characteristic. The finding that these brainstem regions generate and maintain firing at such frequencies is surprising given that gamma band activity was first described in the cortex and associated with such complex processes as consciousness, learning, and memory [3]. Notwithstanding, gamma band activity in the RAS is putatively linked with the process of preconscious awareness and the provision of information that is essential for the formulation of many of our actions and sensations [4,5]. Given its central import, it is no wonder that RAS dysregulation is implicated in many neurological and neuropsychiatric disorders [4]. Herein, we describe recent findings that reveal a potential mechanism for the developmental decrease in REM sleep and the basic rest-activity cycle (BRAC) before addressing the role of gamma band activity in the process of preconscious awareness.

## 2. Gamma Band Activity

Seminal studies reported the presence of PPN neurons *in vivo* that fired at gamma frequencies. Extracellular recordings of PPN neurons *in vivo* identified six categories of thalamic projecting PPN cells according to their firing properties relative to ponto-geniculo-occipital (PGO) wave generation during REM sleep [6]. Some of these neurons had low rates of spontaneous firing (<10 Hz), but most had high rates of tonic firing (20–80 Hz) *in vivo*. Groups of neurons that increased their net firing rate in the PPN during REM sleep were subcategorized as "REM-on". Groups that increased their net firing rate during both waking and REM sleep were subcategorized as "wake/REM-on". Finally, those that increased their net firing rates while waking but that decreased firing during slow-wave sleep (SWS) [7–10] were subcategorized as "wake-on". Altogether, study of these neurons suggested a net increase in firing rates during activated (*i.e.*, waking and REM) states *in vivo.* However, most *in vivo* studies have used "quiet waking" preparations that do not begin to activate PPN cells as they would under a continuous barrage of sensory and motor input, so that the potential firing levels of these cells have not been properly explored. By not stimulating or allowing the animal to move while recording, the firing frequency of PPN cells is limited by the experimenter to a marginally awake (~10 Hz) preparation.

More recently, we described the presence of intrinsic gamma band oscillations in the PPN [1,2]. When every PPN neuron was depolarized, the maximal firing frequency, regardless of transmitter type, was in the beta/gamma range [1,2]. The basal firing rate of most PPN cells without such activation was in the 8–10 Hz range, but higher frequencies (20–60 Hz) were observed as the temperature was raised to physiological levels [4,5]. That is, most *in vitro* recordings are performed at 30–32 °C, basically the body temperature of a hibernating (deeply asleep) animal, but higher frequency activity can be observed by recording at 37 °C or by stimulating the cell using cholinergic agonists [1,2,4,5]. Briefly, all PPN neurons were found to oscillate at beta/gamma frequencies through high-threshold voltage-dependent P/Q- and N-type $Ca^{2+}$ channels. This is the only property shared by all PPN cells regardless of transmitter type [4]. We reported that these $Ca^{2+}$ channels were located on PPN dendrites and alternated in synchrony with membrane oscillations [11]. We also determined that the maintenance of gamma band activity was modulated by G-proteins [12]. We described the presence of gamma band activity mediated by N- and P/Q-type calcium channels in an ascending target of the PPN in the intralaminar thalamus,

the parafascicular nucleus (Pf) [13], and the presence of sodium-dependent subthreshold oscillations in a descending target of the PPN, the subcoeruleus dorsalis (SubCD) [14].

Given that N- and P/Q-type calcium channels contribute to the properties underlying gamma band oscillations in wake-sleep circuits, it becomes imperative to understand their basic mechanisms of action, changing patterns of distribution throughout development, and relation to phenotype. Localization studies have shown the P/Q-channel type is pervasively distributed throughout the brain [15,16], and present throughout the lifespan in rodents [17]. In contrast, the distribution of N-type $Ca^{2+}$ channels often peaks during the early developmental period in rodents [18], an event that presages their gradual replacement with P/Q channels in some regions [19]. Interestingly, altered P/Q- *vs.* N-type $Ca^{2+}$ channel expression manifests markedly different phenotypes [20]. P/Q-type ($Ca_v2.1$) knockout animals exhibit deficient gamma band activity in the EEG, atypical sleep-wake states, ataxic movements, increased risk for seizures (low-frequency synchrony), and death by three weeks of age [21,22]. On the other hand, N-type ($Ca_v2.2$) knockout animals demonstrate few sleep-wake abnormalities but exhibit decreased nociceptive responses and are otherwise normal [20].

Predicated upon the knowledge that P/Q and N-type $Ca^{2+}$ channels might differentially contribute to the gamma oscillations that are vital for waking and REM sleep, we characterized their distribution within the PPN. We reported the presence of three groups of cells based on the expression of N-type channels only (30%, we proposed these as "REM-on"), P/Q-type channel only (20%, we proposed these as "Wake-on"), and both N- and P/Q-type channels (50%, we proposed these as "Wake/REM-on") [23]. These findings supported the notion that different populations of PPN cells generate gamma band activity while waking *vs.* during REM sleep. Indeed, injections of glutamate into the PPN of the rat increased both waking and REM sleep, but injections of NMDA increased only waking, while injections of kainic acid (KA) increased only REM sleep [24–26]. Thus, the two states appear to be independently activated by NMDA *vs.* KA receptors. Also, the intracellular pathways mediating the two states are different. Specifically, KN-93 (a CaMKII activation inhibitor) microinjected into the PPN of freely moving rats decreased waking but not REM sleep [27]. We showed that beta/gamma band oscillations in PPN neurons recorded *in vitro* were blocked by superfusion of KN-93 [28], suggesting that some cells manifest their oscillations via the CaMKII pathway. Figure 1 shows the protocol for identifying N + P/Q, N-only, and P/Q-only cells, and a diagram of the two intracellular pathways modulating N- *vs.* P/Q-type channels. Moreover, we demonstrated that modafinil stimulated increases in electrical coupling via the CaMKII pathway since KN-93 inhibited the action of modafinil [2,28].

We previously demonstrated that G-proteins are vitally important to pedunculopontine processes that are involved in sleep [29]. Others have demonstrated that $G_{oi}$ proteins modulate P/Q-type channels whereas $G_s$ proteins modulate N-type channels [30]. Consequently, intracellular protein kinase C (PKC) is integral to N-type channel activity [31], whereas CaMKII is integral to P/Q-type channel function [32]. That is, the two $Ca^{2+}$ channel subtypes are modulated by different intracellular pathways: N-type by the cAMP/PK pathway and P/Q-type by the CaMKII pathway. Extending these studies further in light of our own recent findings, we hypothesize that there is a "waking" pathway mediated by CaMKII and P/Q-type channels and a "REM sleep" pathway mediated by cAMP/PK and N-type channels, and that gamma band activity is fundamentally different during waking compared to during REM sleep [23]. The distribution of N- *vs.* P/Q-type channels according to PPN electrophysiological type is described in

Figure 2A. The implications of this partitioning are important for understanding the development of wake/sleep cycles, as we will see below.

A separation of cell types by N- *vs.* P/Q-type channel is present in other regions such as the hippocampus. Parvalbumin-expressing hippocampal basket cells manifest P/Q-type channels while cholecystokinin-bearing cells exhibit only N-type channels [33–37], but it is not clear if the two types of basket cells are differentially modulated during specific behavioral states [38]. However, it is known that both channel types are active during different phases of theta oscillations [39]. Gamma band activity, usually requiring activation by cholinergic input, in the hippocampus was divided into fast (>65 Hz) and slow (~25–60 Hz) frequencies that are differentially generated in the CA1 and CA3 subfields, respectively [40]. Such differences have been proposed to "bind" CA1 fast gamma oscillations with very high frequency activity from entorhinal cortex (in charge of providing information about object and place recognition in rodents [41]), whereas CA1 slow gamma oscillations would be locked to the slower frequencies present in the CA3 area in charge of memory storage [40,42]. It is not yet clear if all cells bearing P/Q-type channels fire at different frequencies and in different states compared to cells that manifest N-type channels.

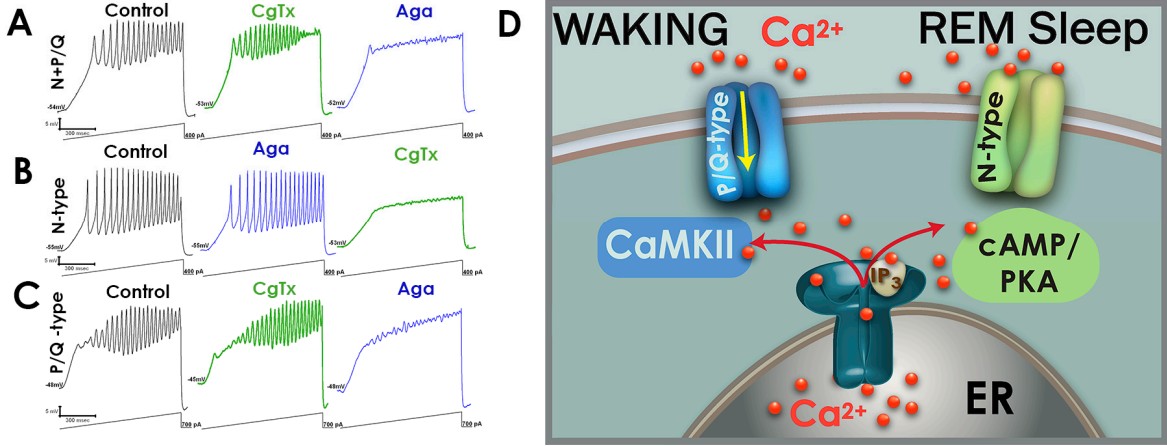

**Figure 1.** Calcium channel PPN cell types, functional and intracellular pathways. (**A**) PPN neuron recorded in the presence of fast synaptic blockers and tetrodotoxin depolarized by a ramp to induce intrinsic membrane oscillations. If application of conotoxin (CgTx) induced a partial decrease in oscillation amplitude and application of agatoxin (Aga) eliminated the remainder of the oscillations, the cell was assumed to have both channel types and was classified as "N + P/Q"; (**B**) In another PPN neuron that was unaffected by Aga but the oscillations eliminated by CgTx, the cell was assumed to have only N-type channels and was classified as "N-only"; (**C**) If a PPN neuron was unaffected by CgTx but the oscillations were blocked by Aga, it was classified as a "P/Q-only" cell [23]; (**D**) Diagram of the CaMKII intracellular pathway modulating P/Q-type channels that contribute gamma band activity during waking, and the cAMP/PK pathway modulating N-type calcium channels that contribute gamma band activity during REM sleep. Both pathways act through the endoplasmic reticulum (ER) and the IP3 mechanism to release intracellular calcium [4].

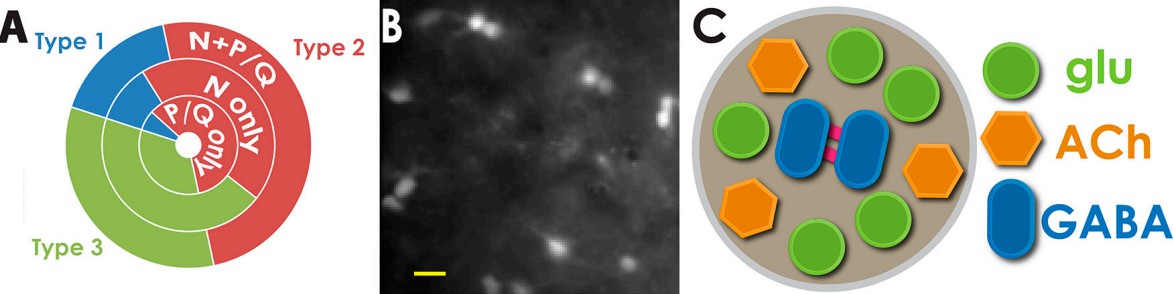

**Figure 2.** Calcium channel PPN cell types, electrical coupling, and ensembles. (**A**) Pie chart showing the distribution of N + P/Q, N-only, and P/Q-only PPN cells according to electrophysiological type. Since type I PPN cells are non-cholinergic, type II are 2/3 cholinergic, and type III cells 1/3 cholinergic, it suggests that PPN neurons with one or both calcium channel types are represented in all transmitter types, cholinergic, glutamatergic, and GABAergic. That is, there are groups of PPN neurons that are N + P/Q that manifest all three transmitter types, as do groups of N-only and P/Q-only cells; (**B**) View of background activity in a PPN slice after loading with Fura2. Note that the majority of cells showing high levels of calcium in the absence of stimulation were found in pairs spaced about 100 μm apart. This suggests the presence of cell ensembles anchored by pairs of electrically coupled cells that are usually GABAergic. Calibration bar 25 μm; (**C**) Glutamatergic neurons are shown as green circles, cholinergic neurons as orange hexagons, and GABAergic cells as blue ovals, some being electrically coupled (red bars). Groups of cells with a 5:3:2 ratio would all need to express the same channel subtype. Ensembles expressing only N-type channels would fire during REM sleep (REM-on), those expressing only P/Q-type channels would fire during waking (Wake-on), and those expressing both would fire during both states (Wake/REM-on) [4].

## 3. Electrical Coupling and Cell Ensembles

We described the presence of dye in electrically coupled pairs of neurons in the RAS, particularly in the Pf, PPN and SubCD [43,44]. Following application of modafinil to PPN and SubCD neurons we observed a decrease in resistance [43,44], commensurate with results found in the cortex, reticular thalamus, and inferior olive [45]. The effects of modafinil were evident in the absence of changes in resting membrane potential or changes in the amplitude of induced excitatory post-synaptic currents, and were inhibited by low concentrations of mefloquine (a gap junction blocker), and also in the absence of changes in resting membrane potential or amplitude of excitatory post-synaptic currents. Together, these studies suggest that these compounds do not act indirectly by affecting voltage-sensitive channels such as $K^+$ channels, but rather modulate electrical coupling via gap junctions. Notably, the electrically coupled cells in these nuclei were almost always GABAergic [43,44]. We also found that the gap junctions responsible for coupling in all three nuclei were composed of connexin 36 and that levels of connexin 36 decrease during development, suggesting that this gap junction protein may participate in developmental regulation and contribute to wake-sleep control in the adult [43].

Anatomically, neurons in the PPN are scattered such that in the *pars compacta* there are glutamatergic (GLU), cholinergic (ACh), and GABAergic neurons in the ratio of 5:3:2, respectively [46]. Studies using

$Ca^{2+}$ imaging in the PPN *pars compacta* reveal an interesting anatomical organization within the nucleus. Figure 2B shows that pairs of PPN cells are labeled throughout the nucleus even in the control, unstimulated condition. The spatial segregation between couplets suggests that there are functional clusters of cells interspersed throughout the nucleus. Since electrically coupled neurons represent GABAergic neurons, we speculate that there are 5 GLU and 3 ACh neurons closely associated with each GABAergic pair. That is, there may be clusters of approximately 10 neurons scattered within the *pars compacta* that create a functional subgroup (Figure 2C). Clearly, additional evidence is required to substantiate this hypothesis. Nevertheless, such is warranted given the possibility that dissecting the organization may elucidate how the entire nucleus generates coherent activity at defined frequencies. Moreover, it is imperative to determine the response of PPN neurons to sensory input and how the response generates coherent activity. This type of functional clustering has been proposed for the hippocampus, particularly in epileptic networks [47]. Cell ensembles were described in the hippocampus as subsets of about 10 neurons that showed repeated synchronous firing during open-field exploration [48]. Interestingly, the timescale of activity between these neurons had a median of 23 ms and the peak optimal timescale was ~16 ms. That is, most activities occurred in the 40–60 Hz range. We hypothesize that a similar temporal relationship will be evident among cell clusters in the PPN.

## 4. Development of REM Sleep

The newborn human exhibits a balanced distribution of waking, rapid eye movement (REM) sleep, and slow wave sleep (SWS), spending about 8 h in each state [49]. After birth, there is a gradual decrease in REM sleep from about 8 h during the perinatal period to about 1 h by age 15, beyond which there is a mild, persistent decrease with aging [49]. During this early developmental period SWS may transiently increase and then gradually decrease from 8 h per day to 6–7 h per day by age 15. The net gain in total waking time, from ~8 h perinatally to ~16 h during adulthood, occurs at the sacrifice of REM sleep duration. Some have suggested that the abundance of REM sleep in development serves to direct the course of brain maturation [50]. This is in keeping with the notion that activity-dependent development may direct neural connectivity throughout the brain [50,51]. Accordingly, it seems plausible that REM sleep could provide endogenous activation at a time when, because of prolonged periods spent sleeping, the brain experiences relatively little exogenous sensory input in comparison to periods in development when greater proportions of time are spent in waking.

In the rat, the decrease in REM sleep occurs between 10 and 30 days of age, declining from more than 75% of total sleep time at birth to about 15% of sleep time by 30 days of age [52]. To determine the origin of the developmental decrease, we explored the changes in transmitter effects on the pedunculopontine nucleus (PPN) during this period [53]. The PPN is critical for the study of REM sleep since it is the most active of the reticular activating system (RAS) cell groups during REM sleep, while the locus coeruleus and raphe nuclei are much less active [54]. We found that some transmitters increased their excitatory (kainic acid) or inhibitory (GABAa, muscarinic, serotonin) actions, while others decreased their effects (NMDA, GABAb, alpha adrenergic) [53]. However, it was difficult to ascribe to any one transmitter the responsibility for the developmental decrease in REM sleep.

We hypothesized that the developmental decrease in REM sleep may be related to the expression of N-type channels instead of P/Q-type channels. We analyzed the expression of the two channel types at

the beginning of the developmental period, 10 days, and at the end, 30 days. We found that there is a decrease of ~350% in the expression of N-type channels in the PPN between 10 and 30 days, while P/Q-type channels showed a modest decrease (~25%) [55]. We compared the levels of expression of the two channels in the hippocampus, which is not known to be related to this developmental change, and found that there was no change in N-type channel, and a small increase in P/Q-type channel, expression in this nucleus. These findings suggest that the developmental decrease in REM sleep may be due at least in part to a decrease in the expression of N-type channels in the PPN (Figure 3).

Another implication of the developmental decrease in N-type channel expression over the developmental decrease in REM sleep is that there may be a reduction of PPN cells bearing only N-type channels. That is, ~30% of PPN cells have N-only channels while ~20% have only P/Q-type channels for cells recorded at ~10 days [23]. By 30 days, there may be a decrease in N-only cells such that there may be equal numbers (or less) of N-only *vs.* P/Q-only cells.

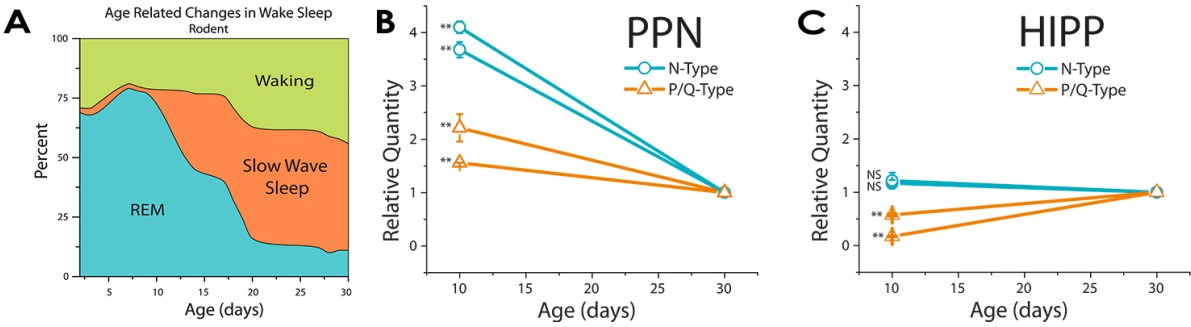

**Figure 3.** The developmental decrease in REM sleep and calcium channel expression. (**A**) Developmental decrease in REM sleep as a percentage of sleep time (after [52]). In the rodent, the decrease occurs between 10 and 30 days before assuming adult levels; (**B**) Relative expression of N-type calcium channels (red lines) and P/Q-type calcium channels (blue lines) at 10 *vs.* 30 days in punches from brain slices containing the PPN. Three replications in each of double samples showed that N-type channel expression significantly decreased >75%, while P/Q-type channel expression decreased <30%; (**C**) Relative expression of N-type calcium channels (red lines) and P/Q-type calcium channels (blue lines) at 10 *vs.* 30 days in punches from brain slices containing the hippocampus (HIPP). N-type calcium channel (red lines) expression did not change between 10 and 30 days, while P/Q-type calcium channel (blue lines) expression increased ~50% during the same period in the HIPP. Results suggest a PPN-specific decrease in N-type channel expression during the developmental decrease in REM sleep [4].

## 5. Basic Rest-Activity Cycle

Classic studies by Kleitman [56] first proposed that the periodic recurrence of the waking *vs.* REM state during sleep reflects the operation of a "basic rest-activity cycle" (BRAC). The influence of the BRAC is thought to modulate brain function during wakefulness via an ultradian rhythm. His studies showed that the sleep-wake cycle of the human neonate was marked by periods of increased activation every 50–60 min. This periodicity gradually increased with age to the ~90 min cycle of the adult, although it was masked by surges of cortical activity during waking.

A substantial volume of evidence has since corroborated the presence of the BRAC during sleep [57], during waking [58], and even in narcolepsy [59]. Early studies recorded gross body movements in adults and found that subjects showed discrete periods of spontaneous activity every 90–120 min during both sleep and wakefulness (reviewed in [60]). Others demonstrated a similar periodicity in eye movements, muscle tone (electromyographic—EMG), electroencephalographic patterns [61], physiological (especially temperature) and behavioral indices [62], errors in continuous performance tasks [63], performance of verbal and spatial tasks [64], and a host of other measures ([65–69]; see [70] for caveats). More recent studies have offered evidence that ultradian oscillations in delta-wave activity in the brain, adrenocorticotropic activity (cortisol secretion), and autonomic activity (heart-rate variability) are coupled to 90–110 min periods [71].

BRACs have been observed in other species also, including primates (REM sleep and EMG cycles around 66 min [72]), felines (REM sleep [73–75] and operant performance [76] cycles around 25 min), and rodents (REM sleep cycles from 8 to 17 min [77–79]). Among animals, there is a linear relationship between the log of the body weight and the log of the period or amplitude of arousal measures [80].

Historically, the BRAC was conceptualized as a mechanism for maintaining homeostasis in the energetically demanding environment of the brain [81–83]. With time, this concept was expanded to encompass a number of integrative functions for diverse metabolic systems under the coordination of basic core brainstem nuclei (central pacemakers) and regulatory factors. Admittedly, many of these highly nuanced dynamics and their linkages with the BRAC remain enigmatic. Nevertheless, recent work provides compelling evidence that glial cells, neurotransmitters, hormones, $Ca_v2$ channels, and intracellular signaling pathways contribute to this rhythm.

Astrocytic oscillations resonate throughout the thalamocortical neural network and may contribute to the BRAC by promoting periods of energy expenditure and rest. On the one hand, astrocytic oscillations regulate neuronal networks by providing energetic support for surrounding neurons during periods of high activity [84–86] via release of ATP through hemichannels [87] and vesicle-dependent mechanisms [88]. On the other hand, astrocyte oscillations inhibit cholinergic basal forebrain neuronal activity to decrease wakefulness and promote periods of rest [89,90]. Given that cholinergic basal forebrain neurons project to the cortex, hippocampus, and thalamus [91], and promote gamma frequencies that underlie perceptual binding during wakefulness [92–94], it seems plausible that astrocytic-induced alterations in cholinergic activity can regulate a functionally important portion of that arousal system by altering ACh levels in the cortex and effectuating alterations in gamma band activity that alter cognitive function. A major question that remains to be answered is, are these astrocytic rhythms in the current frequency range, *i.e.*, ~90 min.

Results from our labs demonstrate that the sleep state-dependent P50 potential in humans, which has been proposed to be an expression of ascending cholinergic activation of the intralaminar thalamus (reviewed in [95]), shows a ~90 min cycle in the peak amplitude of this "preattentional" measure [96]. The rodent equivalent of the human P50 potential, the P13 potential, also was found to show a periodicity in peak amplitude, but at a 13–16 min period, similar to the rodent REM sleep cycle duration. Additionally, human performance on a psychomotor vigilance task (a reaction-time task) showed a 90 min cycle in the lowest number of lapses [96]. This rhythmic waxing and waning of arousal-related activity, therefore, has been more closely tied to a ~90 min frequency.

Also, the BRAC has been related to pulsatile luteinizing hormone (LH) secretion [97]. LH is rhythmically secreted throughout the 24-h cycle in agonadal animals, a trend that co-occurs in phase with BRAC in a number of species [98–100]. In humans, medically-induced ovariectomy results in recurring pulsatile secretion of LH every 75–120 min [99] in keeping with the 90 min adult BRAC cycle. Pulsatile secretion of LH in ovariectomized monkeys recurs in 60–90 min phases that parallel REM and NREM cycles [101]. In ovariectomized cats, LH secretion recurs every 20–30 min and parallels the 25 min REM/non-REM cycle [97]. Interestingly, the onset of puberty effectuates an increase in pulsatile LH secretion [102], suggesting a developmental shift wherein hypothalamic neurons that secrete GnRH may be entrained by BRAC or, alternatively, may mask BRAC rhythms. Regardless, the interspecies intersection of pulsatile LH secretion with measures of BRAC offers credence to the supposition that the two phenomena are related.

Similar pulsatility has been observed in ghrelin, leptin, and insulin, although the latter trend may reflect reciprocal interactions with environmental cues superimposed on the central influence of the BRAC (see [102]). Kleitman [55,103] originally described alternations between active feeding and inactive states in infants placed on a self-demand feeding schedule [55,104], a trend recapitulated in rats fed *ad libitum* [105–107]. Subsequently it was demonstrated that rises in body temperature preceded feedings by 15 min [108]. In rats, these rises were paralleled by episodic (every 1–2 h) increases in brown adipose tissue temperature, a trend that recurred during the 12-h dark and the 12-h light phases [109,110]. The increase in heat production persisted until the end of food consumption, which initiated an abrupt decrease in temperature until baseline values were approximated [111]. This coincident elevation in core temperature may facilitate complex cognitive and synaptic processing that is necessary to actively engage with the external environment during foraging activities since synaptic processing is temperature sensitive [111].

Moreover, peripheral cues that entrain hormonal signals are likely involved. Circulating ghrelin, an orexigenic hormone that induces appetite and feeding behavior, increases before and decreases after every meal [112–116]. Ghrelin levels also appear to modulate insulin and glucagon release [103,117]. Leptin has opposing effects on hunger, ultimately suppressing appetite [118–120], and rising following food consumption. Interestingly, emerging evidence suggests BRAC-related feeding and hormonal release patterns may modulate wider cognitive function [121] and arousal states [29,122]. It has been shown that insulin and ghrelin modulate performance on avoidance learning and spatial memory tasks, both hippocampal-dependent tasks (See [122]). We recently found that leptin decreased sodium currents in all PPN neurons, suggesting that its rise during slow wave sleep may help reduce arousal. Conversely, its decrease during waking is permissive of higher firing frequencies in PPN cells [29,122].

Another implication from the differential control of waking and REM sleep by different $Ca^{2+}$ channel subtypes is the possibility that the BRAC is mediated by the alternation of N- and P/Q-type channels and/or their respective intracellular pathways. Activation of P/Q- and N-type channels in the PPN increases voltage-gated $Ca^{2+}$ conductance and thereby triggers a transient increase in intracellular $Ca^{2+}$ concentration. In turn, $Ca^{2+}$-dependent $K^+$ conductance (via Kv1.1, Kv1.2, and Kv1.6) becomes activated. Following activation, voltage-dependent and voltage-independent regulatory mechanisms are initiated. Voltage-dependent mechanisms necessarily involve G-protein coupled muscarinic receptors, especially M2, the most abundant receptor type in the PPN [123]. The M2 presynaptic receptors bind ACh and inhibit P/Q- and N-type $Ca^{2+}$ currents via feedback mechanisms. That is, acute exposure of

ACh to its cognate receptor results in binding to and activation of the G-coupled M2 protein complex, resulting in exchange of Gα-bound GDP for cytoplasmic GTP and liberation of the βγ subunit. Direct binding of the Gβγ subunit to the pore-forming α1 subunit of the $Ca^{2+}$ ion channel [124–128] induces an inhibited state, requiring stronger depolarization to open [127–130]. The positive shift in voltage-dependent activation in the inhibited state can be reversed following strong depolarization [131,132], an event that displaces the βγ subunit [133,134]. While this type of inhibition has long been established for N-type channels [124,125], more recent work has localized these mechanisms to P/Q-channel function, albeit with a lower degree of inhibition relative to that incurred by N-type channels [135]. Again, it remains to be determined if this reciprocal activity has a ~90 min cycle.

In contrast to the single inhibition pathway induced by direct voltage-dependent interactions between Gβγ subunits and ion channels, there appear to be multiple pathways that induce voltage-independent inhibition of neuronal $Ca^{2+}$ current via a number of second messengers and protein kinases [127,136,137]. For example, termination of signaling can be effectuated by intrinsic GTPase activity of the Gα subunit that induces reassociation of the Gα-GDP subunit with Gβγ [138,139], a process that can be accelerated by RGS proteins (regulator of G protein signaling) [140]. Moreover, intrinsic desensitization mechanisms that alter their response following tonic exposure to agonists can terminate signaling. These mechanisms are complex and involve phosphorylation by PKA, PKC, or G-protein-coupled receptor kinases (GRK), and uncoupling of the G-proteins from their receptor [141]. Others have shown that the presence of Gβγ subunits activates GRKs and induces their translocation from the cytoplasm to the plasma membrane where they phosphorylate G-proteins [142]. G-protein phosphorylation promotes the binding of arrestins and blocks G-protein receptor coupling [143]. Alternatively, tonic exposure to ACh can induce reversible or irreversible internalization of the receptor [142]. These mechanisms result in termination of the GPCR-mediated signaling and, ultimately, enable the P/Q- and N-type channels to return to their basal, unmodulated state. The key question is whether these dynamics manifest a ~90 min cycle.

Once the P/Q channel switches to the unmodulated state, the increased conductance of $Ca^{2+}$ permits a global rise in intracellular $Ca^{2+}$ levels and thereby initiates the activity of regulatory proteins such as CaMKII [144]. Following accumulation of $Ca^{2+}$ and calmodulin, CaMKII becomes persistently activated for a short period as a result of autophosphorylation. CaMKII autophosphorylation, which blocks cAMP when abundant, eventually becomes inactivated upon dephosphorylation of the threonine 286 residue [145], permitting further cAMP activity [146,147]. Thus, desensitization, internalization, and CAMKII autophosphorylation provide mechanisms for promoting oscillatory activity in P/Q- and N-type channels in the presence of tonic ACh exposure [148]. These time-dependent regulatory mechanisms may ultimately reify as the BRAC and account for the frequency of REM sleep episodes during the night along with periodic changes in alertness during the day. More specifically, we hypothesize that preferential activation of N-type $Ca^{2+}$ channels and associated regulatory mechanisms during the troughs result in decreased vigilance during waking hours and REM sleep during sleeping hours. Conversely, preferential activation of P/Q-type $Ca^{2+}$ channels and associated regulatory mechanisms during peaks of this rhythm result in the activation of enhanced wakefulness.

## 6. Preconscious Awareness

Preconscious awareness refers to a state wherein neural information is processed and available but, in the absence of attentional effort, remains beneath the level of conscious awareness, *i.e.*, it is preconscious [149]. Continual activation of the RAS during waking is paramount to maintaining the state of preconscious awareness through its generation of gamma band activity [4,5,28,149]. Upon waking, the flood of visual, auditory, somatosensory, and other afferent input raises the excitability level of these cells to achieve and maintain beta/gamma frequencies [4]. Mechanistically, we proposed that afferent sensory information arising from collateral activation of the RAS triggers dendritic oscillations on PPN neurons. Figure 4 illustrates the pathways involved. Specifically, the presence of sensory inputs to dendritic P/Q- and N-type $Ca^{2+}$ channels induces gamma band oscillations and, in turn, influences overall frequency of PPN firing. The resultant output from the PPN projects inferiorly to the SubCD [150,151], presumably activating neurons that have sodium-dependent subthreshold gamma oscillations. Thereby, these cells affect downstream systems that modulate the atonia of REM sleep and ascending systems that are putatively involved in memory consolidation, particularly in the hippocampus. PPN output also projects superiorly to the intralaminar thalamus, especially the parafascicular nucleus (Pf), inducing oscillations of the dendrites at gamma frequency via P/Q- and N-type $Ca^{2+}$ channels. These cells thereby project to the cortex, particularly to upper cortical layers where the nonspecific thalamic inputs terminate and activate cortical neurons. The resultant background of activity emanates throughout the RAS to create a "context" for sensory experience, while cortically derived input from specific thalamic relay nuclei to layer IV creates the "content" of sensory experience. It is the coincident firing of these two pathways that has been proposed to provide the mechanisms underlying sensory perception [152]. Once gamma band has been generated, it is easily maintained in cortical, hippocampal, and cerebellar cells [153].

Based on the fact that every studied neuron in each of the three major nuclei modulating waking and REM sleep manifests gamma band activity, the RAS was referred to as a "gamma-making machine" [4,5,28,149,154]. We also learned that the various regions of the central nervous system that exhibit gamma band activity—such as the cortex, thalamus, cerebellum, basal ganglia, hippocampus, and RAS—all manifest coherence between regions. That is, these gamma band generators are not isolated but correlated, and in some cases subcortical oscillations precede cortical oscillations. Based on the presence of electrical coupling, intrinsic membrane properties, and circuitry capable of generating and maintaining gamma band activity, we proposed a novel role for gamma band activity in the RAS. While the usual role for gamma band activity in the cortex is that of sensory or motor binding, we hypothesized that continual activation of the RAS during waking manifests gamma activity that is necessary to support a background state capable of reliably assessing the ambient environment [4,5,28,149,154]. Based on our results, we suggested that a mechanism similar to that found in the cortex for achieving temporal coherence at high frequencies is present in the PPN and its subcortical targets (e.g., Pf and SubCD nuclei). We also suggested that gamma band activity and electrical coupling generated in the PPN might stabilize the coherence that is essential for arousal by providing a stable activation state. We identified the intracellular mechanisms that allow this generation and maintenance for prolonged periods along with the intrinsic membrane properties. Thus, we proposed that sensory input induces gamma band activity in the RAS and participates in preconscious awareness (Figure 4).

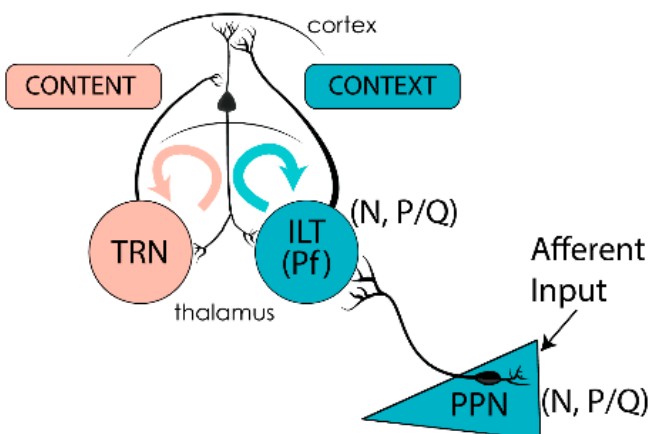

**Figure 4.** Gamma band activity in the RAS and ascending targets. Afferent sensory information impinging on dendrites with high threshold calcium channels induces gamma band activity in PPN neurons. PPN output during REM sleep (N + P/Q and N-only cells) activate SubCD neurons with sodium-dependent subthreshold oscillations (STO), which in turn travel to the hippocampus and descending targets. PPN output during waking (N + P/Q and P/Q-only cells) activate the dendrites of parafascicular (Pf) cells in the intralaminar thalamus (ILT) that also bear high threshold calcium channels in the dendrites [4,5]. Gamma band activity from the "non-specific" ILT travels to upper layers of the cortex to supply the "CONTEXT" of sensory input, while "specific" thalamic relay neurons (TRN) send information to layer IV of the cortex to provide the "CONTENT" of sensory experience. The resulting thalamocortical oscillations based on coincident firing provides conscious perception [153].

William James proposed that the "stream of consciousness" is "a river flowing forever through a man's conscious waking hours" [154]. This pulsing stream persistently infiltrates our essence while waking and yet we often fail to pay it heed, letting much sensory information go unnoticed within the innermost confines of our mind. Once beckoned into awareness, we can actively pay tribute to a particular piece of sensory information: we become fully "conscious" of the information [148]. At the wellspring of this process is the RAS, a phylogenetically conserved area of the brain inundated by the continuous flow of internal and external information. By traversing the RAS, this information modulates wake-sleep cycles, the startle response, and fight *vs.* flight responses (e.g., changes in muscle tone and locomotion). Accordingly, we speculate that activation of the RAS during waking induces coherent activity (through electrically coupled cells) and high-frequency oscillations (through N- and P/Q-type $Ca^{2+}$ channel and subthreshold oscillations) to sustain gamma activity (through activation of G-proteins) and supports a persistent, reliable state for assessing our world.

## 7. Conclusions

Given the foregoing, the functions affected by PPN dysregulation are numerous and involve a multiplicity of processes including perception and response to goal-oriented behavior, voluntary movements, and just about anything else of which we are aware but not explicitly attending. Because of these widespread functions, it is not surprising that several neurological and psychiatric disorders include

PPN dysregulation and, given that the PPN is part of the RAS, necessarily involve wake-sleep problems. Indeed, dysregulation of the wake-sleep cycle may presage the onset of neurological and neuropsychiatric disorders by years, suggesting a window of opportunity for therapeutic intervention. Taken together, all this means that our "old" (in evolutionary terms) brainstem, our primordial emotional survival system, has much more to do with who and what we are than previously thought. The RAS not only modulates waking and arousal but sets the "context" on which we assess the world around us and provides the background upon which we base our movements. Recognition of the aforementioned makes it imperative to better understand the physiological mechanisms that underlie the inner workings of this system so that the development of pharmaceuticals to treat neurological and neuropsychiatric conditions that affect the wake-sleep cycle.

## Acknowledgments

This work was supported by NIH award R01 NS020246, and by core facilities of the Center for Translational Neuroscience supported by NIH award P20 GM103425 and P30 GM110702 to Garcia-Rill.

## Author Contributions

Edgar Garcia-Rill, Brennon Luster, Susan Mahaffey, Stasia M. D'Onofrio, Melanie MacNicol, James R. Hyde, and Cristy Phillips were responsible for the manuscript writing; Garcia-Rill and Luster were responsible for "Introduction"; Garcia-Rill and D'Onofrio were responsible for "Gamma Band Activity"; Garcia-Rill and Hyde were responsible for "Electrical Coupling and Cell Ensembles"; Garcia-Rill, MacNicol and Mahaffey were responsible for "Development of REM Sleep"; Garcia-Rill and Phillips were responsible for coordination of "Basic Rest Activity Cycle"; Garcia-Rill was responsible for "Preconscious Awareness"; Garcia-Rill was responsible for "Conclusions".

## Conflicts of Interest

The authors declare no conflict of interest.

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
