# Peer review of "Pedunculopontine Gamma Band Activity and Development"

_brainsci, doi:10.3390/brainsci5040546_

Round 1

Reviewer 1 Report

This is an outstanding, comprehensive review of the PPN and the RAS and their role in development across the life cycle. The findings on the role of the N-type channels in the PPN (and not in the hippocampus) in the process of developmental decrease of REM sleep represent a notable advance. Further elucidation of the mechanisms subserving the multiple roles played by the BRAC, in particular the astrocytic oscillations resonating throughout the thalamocortical neural network, is impressively documented. The intriguing and provocative (and most likely accurate) culmination of this review relates to Preconscious Awareness and the roles played by the PPN and its gamma band activity and electrical coupling. The authors link their modern scientific findings to the long-ago observations of William James and his proposed "stream of consciousness." The general appeal of this review across a broad range of scientific disciplines cannot be overestimated.

One minor question: page 11, line 387: is reference #148 correct in regards to preconscious awareness? That reference is a review of protein kinase, from form to function. Reference #149 is entitled "The preconscious and potential space."

Author Response

Thank you for the review, the correction has been made.

Reviewer 2 Report

  This is an important review of work from the authors’ laboratory to date on the role of the gamma-band generating neurons in the PPN area of the reticular activating system. In particular, their work has importantly expanded our understanding of the role of the P/Q-type Ca2+ channels in the control of Waking and the N-type Ca2+ channels in the control of Rapid Eye-Movement sleep (REMS).  Furthermore, the observations on the development of this system in brainstem may help explain the well documented maturational decrease in REMS that has been reported for nearly every species studied.

 Aside from one or two comments and some typos, this review is ready for publication.  I am sure the authors will be able to fix these problems. I list these here:

Page 2 (P2) Line 34, the statement that “every cell” manifest the beta/gamma band oscillations, implies that they have recorded from every cell in the RAS, which is not likely. Perhaps every recorded cell exhibits these patterns, but this is not the same thing.

P5 L138. Figure 2/B. should have a scale bar.

P5 L147 There is a symbol missing from my copy between “100” nad “m apart”.

P7 L233 “in part by a decrease” should be “in part to a decrease”

P11 L370 “CA2+” should be “Ca2+”

P11 L388 “through of its generation” should be “through its generation”

P12 L420 again with the “every neuron”, unless they have recorded from every neuron, this is misleading.

Author Response

Thank you for the wonderful review.

All of the suggestions were made as requested.

Reviewer 3 Report

This is a well-written review describing recent advances in the development of the pedunculopontine nucleus (PPT). Much of the emphasis is on the discovery by this group of intrinsic membrane oscillations that generate gamma frequency activity in PPT cells. The authors cite early literature showing that indeed these cells fire at such frequencies in vivo. This group identified the fact that these membrane oscillations are mediated by high threshold, voltage-dependent calcium channels of the N- and P/Q-types. Recently, they also reported that there are different populations of cells with either or both channel that may mediate gamma activity during waking vs REM sleep. That is, they previously proposed that waking is modulated by P/Q-type channels that are controlled by CaMKII, while REM sleep is modulated by N-type channels controlled by cAMP. These findings are in keeping with considerable work from the Datta lab, who showed that waking is modulated preferentially by CaMKII while REM sleep is modulated by cAMP.  The link to two specific channel types is a significant advance.

This is the basis for their description of a recent report that showed that the expression of N-type channels in the PPT (but not the hippocampus) decreases in conjunction with the developmental decrease in REM sleep. This is another major discovery by this lab, which is outlined in the present review. This mechanism, decreased N-type channel expression, helps explain a phenomenon, the developmental decrease in REM sleep -- that we have known about since the mid-1960s.

The review goes on to describe a number of mechanisms that may mediate the basic rest activity cycle (BRAC), which was first described by Kleitman in the 1950s. Certainly addressing the BRAC is necessary for a review on development, but the sections on astrocytes and the P50 potential could be shortened. Otherwise, this is an excellent review well worth publishing.

I have two very minor suggestions for this manuscript:

1)    Please check numbers of citation for accuracy: Page 10, line 4 (manuscript line 331), should 121 be 122?

2)    Page 11, 6. Preconscious Awareness, lines 10 – 14, “The resultant output from the PPN projects inferiorly …… Thereby, these cells affect downstream ……… memory consolidation, particularly in the hippocampus.” (Need original citations)

For example:  1) Datta S, Li G, Auerbach S. Activation of phasic pontine-wave generator in the rat: a mechanism for expression of plasticity-related genes and proteins in the dorsal hippocampus and amygdala. Eur. J. Neurosci. 27:1876-1892 (2008). 2) Datta S, O'Malley MW. Fear extinction memory consolidation requires potentiation of pontine-wave activity during REM sleep. J. Neurosci. 33:4561-4569 (2013).

Author Response

Thank you for the thoughtful review.

The reference was corrected to 122.

The citations suggested were added.
